# Viral Epitope Scanning Reveals Correlation between Seasonal HCoVs and SARS-CoV-2 Antibody Responses among Cancer and Non-Cancer Patients

**DOI:** 10.3390/v16030448

**Published:** 2024-03-13

**Authors:** Salum J. Lidenge, Dicle Yalcin, Sydney J. Bennett, Owen Ngalamika, Brenda B. Kweyamba, Chacha J. Mwita, For Yue Tso, Julius Mwaiselage, John T. West, Charles Wood

**Affiliations:** 1Department of Clinical Research, Training, and Consultancy, Ocean Road Cancer Institute, Dar es Salaam P.O. Box 3592, Tanzania; sjlidenge@yahoo.co.uk (S.J.L.); bbashekera@yahoo.com (B.B.K.); jmwaiselage@yahoo.com (J.M.); 2Department of Clinical Oncology, Muhimbili University of Health and Allied Sciences, Dar es Salaam P.O. Box 65001, Tanzania; 3Department of Interdisciplinary Oncology, Stanley S. Scott Cancer Center, Louisiana State University Health Sciences Center, New Orleans, LA 70112, USA; dyalci@lsuhsc.edu (D.Y.); sydney.townsend14@huskers.unl.edu (S.J.B.); ftso@lsuhsc.edu (F.Y.T.); jwest6@lsuhsc.edu (J.T.W.); 4School of Biological Sciences, University of Nebraska-Lincoln, Lincoln, NE 68516, USA; 5Dermatology and Venereology Division, University Teaching Hospital, University of Zambia School of Medicine, Lusaka P.O. Box 50001, Zambia; owen_ngalamika@yahoo.com

**Keywords:** SARS-CoV-2, HCoVs, VirScan, antibody, cancer

## Abstract

Seasonal coronaviruses (HCoVs) are known to contribute to cross-reactive antibody (Ab) responses against SARS-CoV-2. While these responses are predictable due to the high homology between SARS-CoV-2 and other CoVs, the impact of these responses on susceptibility to SARS-CoV-2 infection in cancer patients is unclear. To investigate the influence of prior HCoV infection on anti-SARS-CoV-2 Ab responses among COVID-19 asymptomatic individuals with cancer and controls without cancers, we utilized the VirScan technology in which phage immunoprecipitation and sequencing (PhIP-seq) of longitudinal plasma samples was performed to investigate high-resolution (i.e., epitope level) humoral CoV responses. Despite testing positive for anti-SARS-CoV-2 Ab in the plasma, a majority of the participants were asymptomatic for COVID-19 with no prior history of COVID-19 diagnosis. Although the magnitudes of the anti-SARS-CoV-2 Ab responses were lower in individuals with Kaposi sarcoma (KS) compared to non-KS cancer individuals and those without cancer, the HCoV Ab repertoire was similar between individuals with and without cancer independent of age, sex, HIV status, and chemotherapy. The magnitudes of the anti-spike HCoV responses showed a strong positive association with those of the anti-SARS-CoV-2 spike in cancer patients, and only a weak association in non-cancer patients, suggesting that prior infection with HCoVs might play a role in limiting SARS-CoV-2 infection and COVID-19 disease severity.

## 1. Introduction

The coronavirus disease 2019 (COVID-19) is caused by a novel severe acute respiratory syndrome coronavirus type-2 (SARS-CoV-2) [1]. Since the first reported case of COVID-19 at the end of 2019, SARS-CoV-2 has rapidly spread throughout the world [2]. SARS-CoV-2 is a member of the positive-sense RNA virus class, which includes the alphacoronaviruses (α-CoVs) NL63 and 299E, and the betacoronaviruses (β-CoVs) OC43, HKU1, Middle East Respiratory Syndrome-coronavirus (MERS-CoV), SARS-CoV-1 and SARS-CoV-2 [3]. These CoVs express several proteins including structural proteins (spike, envelope, membrane, and nucleocapsid) together with accessory proteins encoded by open reading frames (ORFs) 3, 6, 7, 8, 9, 10, and 14, and non-structural proteins encoded by ORF1 [4]. There is a high degree of homology among HCoVs in terms of both structural and non-structural proteins [5]. The seasonal common cold coronaviruses (HCoVs) OC43, HKU1, NL63, and 299E are common in human populations worldwide, causing mild seasonal respiratory diseases [6,7]. MERS-CoV, SARS-CoV-1, and SARS-CoV-2 can cause a diverse range of symptoms, from asymptomatic to severe or fatal disease [8,9]. The major difference between other CoVs and SARS-CoV-2 infection is the spread by asymptomatic individuals and a more efficient human-to-human transmission [10,11,12,13,14,15].

Because of the high homology between SARS-CoV-2 and other CoVs, and the very high prevalence of seasonal CoVs, it is plausible that these viruses play a role in cross-reactive immune responses, as well as in providing cross-protection against SARS-CoV-2 infection and limiting disease severity [16,17,18,19,20,21]. Whether the source of cross-reactive responses is from seasonal CoVs or other infectious diseases is not clear. In other studies, pre-existing cross-reactive immune responses have been shown to protect against infections such as influenza A and Japanese encephalitis virus infections [22,23]. However, with other infections, like Dengue virus and Zika virus infections, these responses promote disease severity [24,25]. The influence of seasonal CoVs on anti-SARS-CoV-2 and cross-reactive immune responses to SARS-CoV-2 is not well understood. Understanding the role of pre-existing cross-reactive responses may contribute to our understanding of the heterogeneity of clinical presentations and outcomes in COVID-19 disease and vaccination. This is important for designing pan-coronavirus vaccines that can protect from emerging SARS-CoV-2 variants of concern (VoCs) and future novel pandemic CoVs.

Correlates of protection against SARS-CoV-2 infection and COVID-19 disease severity have been associated with demographic factors such as age, male gender, and black and Asian ethnicities [26,27,28,29]. Respiratory diseases, diabetes, cancer, and cardiovascular diseases have also been implicated [30,31,32,33]. Various immunological correlates, like autoantibodies to type I interferons, HIV infection, elevated serum cytokine levels, neutralizing antibodies (nAbs), and SARS-CoV-2-specific CD4^+^ and CD8^+^ T cell responses, have also been associated with COVID-19 severity and disease outcomes [34,35,36]. However, studies on these correlates, particularly in African cancer patients, are lacking. Cancer patients are at a higher risk of SARS-CoV-2 infection and COVID-19 since they tend to be older, are likely to have multiple comorbidities, and are often immunosuppressed due to cancer and cancer treatment [37,38]; yet, less is known about SARS-CoV-2 infection and the humoral responses of cancer patients, especially in sub-Saharan Africa (SSA) where cancer is often associated with other comorbidities like HIV at a relatively younger age. Several reports have indicated that cancer patients have an increased risk of severe COVID-19 with an approximately 3.5-fold increase in the risk of ICU admission, needing mechanical ventilation, or death compared with patients without cancer [39,40,41]. Cancer patients’ increased susceptibility to severe complications of COVID-19 can be due to their compromised immune response caused by the cancers and their treatments. In addition, cancer is associated with an overall blunted immune system, chronic inflammation, and an increase in the possibility of complications [42].

Although the detection of antibodies to SARS-CoV-2 does not necessarily imply protective immunity, recent reports from passive immunizations with convalescent sera and anti-SARS-CoV-2 vaccination indicate that neutralizing antibody (nAb) directed against S protein and the receptor-binding domain (RBD) can prevent infection or mitigate disease [43,44,45,46]. We have also recently reported the presence of nAb and non-neutralizing Ab (antibody-dependent cellular cytotoxicity (ADCC)) responses in the plasma of non-cancer COVID-19 patients [47]; however, the presence and magnitude of these responses in African cancer patients have not been investigated. Similarly, our group and others have reported the presence of cross-reactive anti-HCoV antibodies in non-cancer individuals [16,17,18,19,20,21], but whether infection with HCoV impacts the magnitude or breadth of anti-SARS-CoV-2 responses in cancer patients is unknown.

To study the influence of prior seasonal HCoV infection on anti-SARS-CoV-2 Ab responses among COVID-19 asymptomatic individuals with cancer we utilized the VirScan technology, in which phage immunoprecipitation and sequencing (PhIP-seq) of antigens recognized by antibodies in the individuals with cancer and non-cancer controls were used to investigate high resolution (i.e., epitope level) humoral responses. Correlations between prior history of seasonal CoV infection and the magnitude and breadth of anti-SARS-CoV-2 responses were conducted.

## 2. Materials and Methods

### 2.1. Study Design, Subjects, and Samples

In this longitudinal study, we recruited consenting adults >18 years of age and of both genders from Dar es Salaam, Tanzania during the COVID-19 pandemic and followed them up for up to 12 weeks between August 2021 and March 2022. A total of 62 SARS-CoV-2 seropositive individuals were included in this study at baseline (26 individuals diagnosed with cancer and 36 individuals without cancer diagnosis) and 15 individuals contributed at least one follow-up visit in the 12 weeks of follow-up. The participants with cancer were recruited at the Ocean Road Cancer Institute (ORCI) in Tanzania following a diagnosis of cancer but prior to the cancer-specific treatment. The cancer diagnoses were made by histological examination of biopsy specimens according to the Tanzania standard guidelines for specific cancer diagnosis. The non-cancer participants were mainly individuals escorting cancer patients to the hospital and those attending non-cancer hospital services. Because of the ongoing HIV pandemic in SSA, the prevalence of both AIDS-associated and non-AIDS-associated cancers is high among people living with HIV (PLWH). These were mainly cervical cancers and Kaposi sarcomas, the two major HIV-associated cancers seen at the ORCI. Consequently, there was a higher prevalence of HIV among cancer patients compared to non-cancer individuals.

Following the informed consent procedure, a structured interview was conducted using a questionnaire to collect the sociodemographic and clinical characteristics of the study participants, including age, gender, HIV status, and history of SARS-CoV-2 infection and COVID-19 disease. Approximately 2 ml of blood was collected from each patient for SARS-CoV-2 and HIV serology screening and subsequent study assays. All study procedures were approved by the Tanzania National Institute for Medical Research (NIMR/HQ/R.8a/Vol. IX/3750).

### 2.2. HIV-1 Serological Testing

HIV-1 serology was determined by the HIV Rapid Test Algorithm (United Republic of Tanzania, 2007) in Tanzania [48].

### 2.3. Immunofluorescence Assay against SARS-CoV-2 Spike and Nucleocapsid Proteins

To screen for the presence of antibodies against SARS-CoV-2 among the study participants, we used an in-house immunofluorescence assay (IFA) against the spike and nucleocapsid proteins of SARS-CoV-2, as previously described [16]. Briefly, HEK-293T cells (ATCC, Manassas, VA, USA) were transfected with mammalian expression plasmids encoding either the spike or nucleocapsid proteins of SARS-CoV-2 (Addgene and Sino Biological, Wayne, NJ, USA). After 48 h, the transfected cells were fixed and seeded onto 12-well polytetrafluoroethylene (PTFE) printed slides (Electron Microscopy Sciences, Hatfield, PA, USA) with each well containing either spike, nucleocapsid, or mock transfected cells.

Each plasma sample was diluted at 1:20 with 1X PBS, 0.1% Tween-20, and incubated at room temperature for 30 min. The IFA slides were incubated with 1X PBS, 0.1% Tween-20 for 30 min at 37 °C. The diluted plasma samples were then added to each well and incubated for 30 min at 37 °C, followed by incubation with a secondary mouse monoclonal anti-human IgG antibody (ATCC, USA) for 30 min at 37 °C. Tertiary CY2 conjugated donkey anti-mouse IgG (Jackson ImmunoResearch Laboratories, West Grove, PA, USA) was added and incubated for 30 min at 37 °C. Finally, the slides were counterstained with 0.004% Evans blue solution and then washed. The slides were examined by three independent readers with a Nikon Eclipse 50i fluorescence microscope, and the positive cells appeared green in color. A well was only considered positive or negative if at least two readers reached the same result independently. The samples that were positive for anti-SARS-CoV-2 antibody at 1:20 dilution were further serially 2-fold diluted to determine the anti-SARS-CoV-2 antibody titer. The results were plotted into graphs and statistical analysis was conducted using GraphPad (GraphPad Prism v10, USA).

### 2.4. Pseudovirus Production and SARS-CoV-2 Spike Protein Neutralization Assay

The production of SARS-CoV-2 pseudo-viruses for the neutralization assay was performed as previously described [47,49]. Briefly, SARS-CoV-2 spike glycoprotein pseudotyped lentiviruses were generated by the co-transfection of HEK-293T cells with the SARS-CoV-2 spike mammalian expression plasmid (pcDNA3.1-SARS2-S, a gift from Dr. Fang Li, 145032, Addgene, Watertown, MA, USA), 3rd generation lentiviral plasmid encoding EGFP (FUGW, a gift from Dr. David Baltimore, 14883, Addgene, Watertown, MA, USA), and the packaging plasmid (psPAX2, a gift from Dr. Didier Trono, 12260, Addgene, Watertown, MA, USA). The culture supernatant containing the pseudotyped virus was collected at 72 h post-transfection, concentrated by ultracentrifugation, and stored at −80 °C until assay time.

Twenty-four hours before the neutralization assay, 1.3 × 10^6^ 293T-hACE2 cells/10 mL media were seeded in a Poly-L-Lysine treated 96-well plate. The plasma samples were heat-inactivated at 56 °C for 1 h and diluted at 1:400 with culture medium and 25 µL of the SARS-CoV-2 spike pseudotyped virus for a total volume of 200 µL per well. The plasma–virus mixtures were then incubated at 37 °C for 1 h. The plasma–virus mixture was then added to the HEK-293T-hACE2 cells, spun at 400× *g* for 20 min, and incubated at 37 °C in a 5% CO_2_ incubator for 72 h. The level of infection was determined by quantification of the GFP signal using a BD Accuri C6 Plus flow cytometer (BD Biosciences, San Jose, CA, USA) and flow data analyzed by FlowJo software v10.8.1 (BD Life Sciences, San Jose, CA, USA). Each plasma sample was tested in duplicates. Each set of experiments contained mock-only cells and virus-only cells. The percent GFP in the mock cells was then subtracted from all other samples, including virus-only cells. To calculate the final percent neutralization, the following equation was used: (Virus only − Sample)/Virus only × 100%. The data were then plotted and analyzed using GraphPad Prism v10 (GraphPad Software, San Diego, CA, USA).

### 2.5. Phage Immunoprecipitation and Sequencing (PhIP-Seq)

The T7-HCoV-56-mer phage library was generously provided to us by Dr. Stephen J. Elledge at Harvard Medical School [50]. Our group and others have previously described the phage immunoprecipitation and sequencing (PhIP-Seq) protocol [50,51]. Briefly, the phage library was amplified and titered per the manufacturer’s instructions (Novagen T7Select System, Burlington, MA, USA). Then, 1.4 × 10^9^ pfu of the library (≈2 × 10^5^ pfu/library member) was mixed with plasma containing 2 µg of total IgG. The phage–antibody complexes were immunoprecipitated using Protein A and Protein G magnetic beads (10008D/9D, Invitrogen, Boston, MA, USA) and a magnetic separation rack (S1511S, NEB, Ipswich, MA, USA). After immunoprecipitation, the beads were resuspended in 40 µL of nuclease-free water, heated to 95 °C to lyse the phage, and the phage DNA was PCR amplified. During PCR amplification, each well of the 96-well plate was barcoded, and Illumina adaptors were added. The resulting 376 bp amplicon was then gel-extracted and sent to the Genomics core for quality assessment and sequencing. Clustering and sequencing were performed on an Illumina NextSeq550 (50-cycle) using a mid-output flow-cell single-end read protocol [49,51]. The run was monitored by an Illumina Sequence Analysis Viewer, and the raw FASTQ files were generated following de-multiplexing.

### 2.6. PhIP-seq Data Processing and Statistical Analyses

The PhIP-seq data were analyzed as previously described [49]. The phage library annotation database contains various information, such as the amino acid sequence and taxonomic identification for each isolate from strain to kingdom levels. Each peptide in the database is assigned a unique identifier based on its oligonucleotide sequence, which allows for the consolidation of raw counts from peptides with 100% identity. This prevents misrepresentation of the antibody responses and ensures an accurate assessment of the breadth and magnitude of these responses. To maintain data integrity, peptides associated with obsolete, redundant, or deleted UniProtKB IDs were either updated or removed from downstream analyses. Additionally, the database underwent manual curation to ensure consistent protein names within and across organisms.

The PhIP-stat tool (https://github.com/lasersonlab/phip-stat, accessed on 15 June 2023) was utilized for generating the raw count data, as well as for identifying significantly enriched peptide responses obtained through Gamma–Poisson fitted residual *p*-values for each peptide. −log_10_(*p*), or MLXP, was used to represent the magnitude of the Ab responses, which reflects the frequency with which a peptide is targeted. Peptides with an MLXP > 1.3 (i.e., *p* < 0.05) were called enriched hits and determined the presence/absence of *reactive* peptides, a term that indicates that the plasma had an antibody that reacted against the given peptide. We defined *breadth* as the sum of reactive peptides per protein/proteome per sample. All statistical analyses were performed in R v.4.2.0 or GraphPad Prism 10. Mann–Whitney U tests were conducted to compare groupwise breadth and magnitude of responses. The Wilcoxon matched-pairs signed rank test was used to compare baseline and follow-up Ab responses. Lastly, contingency analyses of demographic parameters were conducted using Fisher’s exact test.

## 3. Results

### 3.1. Sociodemographic and Clinical Characteristics of the Study Participants

At baseline, we included 62 SARS-CoV-2 seropositive individuals with no history of prior confirmed COVID-19 diagnosis. The cohort comprised 26 individuals with cancer and 36 non-cancer individuals. The individuals with cancer were predominantly diagnosed with HIV-associated Kaposi sarcoma (KS; n = 19). The majority of individuals with KS had disseminated cutaneous KS disease (n = 10), followed by localized cutaneous (n = 6), and visceral KS (n = 3). Other cancers included cervical cancer (n = 4), breast cancer (n = 2), and colon cancer (n = 1). The median age of the cohort was 44 years (ranging from 33 to 63 years). However, individuals with cancer were significantly younger compared to non-cancer individuals with median ages of 33 and 55 years, respectively (*p* = 0.003). The cohort was well represented in terms of gender, where there were no significant differences observed (Table 1). There was, however, a significant association between HIV status among cancer and non-cancer individuals (*p* < 0.0001). Of the 25 (~40%) HIV positive-individuals in the cohort, the majority, 19 (73%), had cancer, compared to 6 (16%) without. Although all HIV-positive patients were on ART with reported good adherence, unfortunately, their CD4 counts, plasma HIV viral load, and duration of ART use/HIV infection were not available for this study.

As expected in most SSA countries, the majority (~75%) of the study participants reported a history of TB (Bacillus Calmette–Guérin, BCG) vaccination and this was similar between the cancer and control groups. However, the rate of SARS-CoV-2 vaccination was low at the time of recruitment, only 5 (~8%) participants had received either the AstraZeneca or Janssen vaccines. The majority of the participants, 44 (~71%), were asymptomatic for COVID-19 at the time of recruitment despite testing positive for the anti-SARS-CoV-2 antibody in their plasma (Table 1). Some individuals reported certain non-specific symptoms, such as fever, difficulty in breathing (DIB), cough, or their combinations, detailed in Table 1. Despite these reported symptoms, none of the patients required hospitalization due to COVID-19.

### 3.2. High-Resolution Humoral Antibody Responses against SARS-CoV-2 and HCoVs

To investigate and compare Ab responses against common seasonal human coronaviruses (HCoVs), in addition to anti-SARS-CoV-2 responses, in asymptomatic individuals with and without cancer from SSA, we quantified Ab responses which can be represented as breadth and magnitude (see Section 2). Both the immunoprecipitated phage:Ab complexes from individuals with and without cancer, as well as from the no-plasma controls (mock IP/PBS), were barcoded for multiplexed high-throughput sequencing of epitope-encoding segments. The CoV phage library, generated using the VirScan technology [52,53], comprises 56-mers with 28 amino acid overlaps and includes viral epitopes belonging to SARS-CoV-2, as well as nine other coronaviruses (three Bat coronaviruses (BtCoVs), four common seasonal human coronaviruses (HCoVs), SARS-CoV-1, and MERS-CoV). The input library representations of the SARS-CoV-2 and HCoVs are shown in Appendix A. Replicate summary statistics after sequencing displayed high-quality reads and alignment rates (Appendix A). Furthermore, the 3D-PCA projections based on the presence or absence of significantly enriched (i.e., *reactive*) peptides showed a clear separation between technical controls and biological replicates, following data-preprocessing steps (Appendix A). After data quality assurance, we additionally calculated the organism-level and protein-level breadth (defined as the sum of reactive peptides in each level), the percent subject reactivity against each peptide, and the organism/protein-level average magnitude (represented as the average frequency at which reactive peptides were targeted in each level). These measures allow for the comparison of breadth and magnitude between proteins/organisms within and between groups and the identification of consistently or rarely targeted peptides (public and private epitopes, respectively), in addition to the high-resolution peptide-specific Ab responses. It is important to note that the magnitude, not the breadth, of the SARS-CoV-2 responses was significantly correlated (weak-to-moderate) with the immunofluorescence assay-determined anti-spike and anti-nucleocapsid Ab titers from the same subjects (Appendix A, Spearman *ρ* = 0.34 and 0.54, respectively).

As expected, the SARS-CoV-2 epitopes were more consistently recognized, having a consistently higher breadth of response across most of the individuals compared to other seasonal coronaviruses (Figure 1A). The organism-level breadth of recognition among individuals was not significantly influenced by HIV serostatus (*p* = 0.19), age (*p* = 0.16), or gender (*p* = 0.44). Furthermore, the breadth of responses against the entire CoV repertoire (combined responses of all coronaviruses in the VirScan library) between individuals with and without cancer did not reveal statistically significant differences. When we further stratified the cancer group based on whether they had KS vs other cancers, interestingly, we observed that KS patients had a significantly broader CoV repertoire compared to individuals with other cancers (*p* = 0.0027) (Figure 1B). Similar observations were made when seasonal coronavirus repertoires were distinguished from SARS-CoV-2-specific responses (Figure 1C,D). Consequently, our data also revealed that individuals with other cancers, in contrast to KS, had significantly lower overall CoV recognition compared to non-cancer individuals; however, this difference was not strongly evident, due to large differences in sample size (*p* = 0.037). Taken together, the SARS-CoV-2 epitopes were the most consistently recognized in both cancer and non-cancer individuals compared to seasonal HCoVs, and these responses were independent of age, sex, and HIV status.

In addition to comparing the cancer and non-cancer groups, we also investigated whether treatment had an impact on the CoV repertoire recognized or the magnitude of responses among cancer patients. Both pattern recognition-based hierarchical clustering and statistical comparisons within a subset of paired data of cancer patients who had 12 weeks of follow-up revealed highly patient-specific recognition patterns, regardless of cancer type or treatment regimen (n = 12) (Figure 2A). Similarly, and more prominently, the magnitudes of these responses were also patient-specific and did not segregate baseline from treated cancer patients (Figure 2B).

### 3.3. Magnitude of Antibody Responses in Individuals with and without Cancer

Knowing that all patients were seropositive for SARS-CoV-2 at the time of recruitment, expectedly, SARS-CoV-2 was the most consistently recognized organism in the library by both individuals with and without cancer compared to other seasonal coronaviruses. We, therefore, investigated whether the magnitudes of Ab responses against SARS-CoV-2 or other seasonal coronaviruses-specific antibody responses differed in cancer patients when compared to participants with no disease. Hierarchical clustering revealed that SARS-CoV-2 appeared to have a higher overall magnitude of responses among participants when compared to other HCoVs. However, the average magnitude of SARS-CoV-2 responses was higher (*p* = 0.03) in the majority of individuals without cancer (Figure 3A). It should be noted that these responses were also not significantly influenced by the age (*p* = 0.78), sex (*p* = 0.44), or HIV status of the participants (*p* = 0.17).

We further compared the magnitude of responses between groups, focusing on the protein-level SARS-CoV-2 responses. The magnitude of the SARS-CoV-2 specific antibody responses was consistently higher against spike, nucleocapsid (NC), ORF1, ORF3, ORF8, and membrane proteins (Figure 3B). The anti-NC and anti-ORF4 magnitudes were, on average, significantly higher in non-cancer individuals, while the anti-ORF1 magnitudes were significantly higher in cancer patients (*p* = 0.031, *p* = 0.043, and *p* = 0.040, respectively) (Figure 3C). Similar observations were made when only KS patients were compared to non-cancers, with stronger evidence for NC and ORF1 proteins (*p* = 0.013 and *p* = 0.022, respectively).

### 3.4. Influence of Seasonal Coronavirus Repertoire on SARS-CoV-2 Specific Antibody Responses and Neutralization

In contrast to the organism-level magnitude of responses demonstrating a higher SARS-CoV-2 specific magnitude of responses in non-cancer individuals, the SARS-CoV-2 spike neutralization data clearly showed that cancer patients had significantly higher SARS-CoV-2 neutralization compared to non-cancer individuals. Moreover, amongst the 19 subjects with neutralizing responses above 50% at 1:400 plasma dilution, 18 (95%) were KS patients (Figure 4A). Our group and others have reported the presence of cross-reactive Ab responses against SARS-CoV-2 due to prior infection with common seasonal human coronaviruses [16,17,18,19,20,21]. Cross-reactive epitopes might offer some level of protection against SARS-CoV-2 [20,54,55]. We investigated whether a prior history of HCoV exposures influenced SARS-CoV-2 Ab responses, in addition to comparing SARS-CoV-2 neutralization activity amongst individuals. Since neither anti-S nor anti-NC magnitudes specific to SARS-CoV-2 showed differences between cancer and non-cancer individuals (Figure 3C), and yet anti-HCoV and anti-SARS-CoV-2 had overall higher breadths in KS patients compared to non-KS cancer patients (Figure 1C,D), we looked for associations between HCoV- and SARS-CoV-2-specific anti-spike responses. Both anti-S and anti-NC breadth were significantly higher in KS patients than in patients with other cancers (Figure 4B). In addition, we observed that in cancer patients, specifically in KS patients, there appeared to be a significantly moderate-to-strong correlation between HCoV (HKU1-N2 being the highest contributor, spearman *ρ* = 0.67) and SARS-CoV-2 average anti-spike magnitudes (Figure 4C). This association, although positive, was weaker among non-cancer individuals, where HCoV-spike responses were poorly predictive of SARS-CoV-2 spike responses (Figure 4D). Our data therefore suggest that there is an influence of prior seasonal coronavirus infections on the magnitude of anti-SARS-CoV-2 responses of KS patients.

## 4. Discussion

In this study of Tanzanian individuals with and without cancer, we utilized phage immunoprecipitation and sequencing (PhIP-seq) to investigate the influence of prior infection with seasonal CoVs on the breadth and magnitude of anti-SARS-CoV-2 Ab responses. Consistent with the study design where all recruited participants were SARS-CoV-2 seropositive, we found that the SARS-CoV-2 epitopes were the most consistently recognized in the Ab repertoires. Cancer did result in a differential HCoV Ab repertoire from that in subjects without cancer; however, the magnitude of the anti-SARS-CoV-2 Ab responses was lower in the cancer group, particularly for KS patients. Other variables, such as age, gender, HIV status, and chemotherapy regimen, had no significant influence on the CoV Ab repertoire. Similarly to previous reports, infection with seasonal CoVs appeared to correlate with the magnitude of anti-SARS-CoV-2 Ab responses [18,19,20].

The lack of a differential CoV repertoire, as well as clinical presentation, between cancer and non-cancer participants, was surprising given that almost all the individuals with cancer were also HIV infected. Perhaps the effect of HIV might have been blunted by the fact that all HIV-infected individuals had reported high-level ART adherence. Both cancer status and chemotherapy for cancer treatment did not appear to significantly affect the CoV Ab repertoire either. The observed lower magnitude of anti-SARS-CoV-2 Ab responses among individuals with cancer, although statistically significant, does not appear to have clinical significance because there were no obvious differences at presentation and, significantly, there were no obvious symptoms or disease in both groups. As observed in this study, where seasonal CoV infections appeared to correlate with anti-SARS-CoV-2 responses it is possible that these prior infections induced immune responses that equally benefited individuals with and without cancer in this setting.

Unexpectedly, we also found that cancer patients (KS) had higher levels of SARS-CoV-2 nAb compared to non-cancer controls, even though they had an overall lower anti-SARS-CoV-2 magnitude. Moreover, there were no detectable differences in responses against the spike protein, suggesting that the magnitude of PhIP-seq delineated responses against the SARS-CoV-2 spike does not reflect viral neutralization responses. The underlying reason is unclear, but one possibility is that beneficial immune cross-reactivity in SARS-CoV-2 infection may also be derived from other infectious sources. Cross-reactive responses from unrelated organisms, such as influenza and CMV have been reported [56]. Since KS patients at the ORCI usually present with late-stage disease, they are more likely to be susceptible to multiple co-infections and chronic inflammation. In addition, the KS individuals in our cohort were predominantly co-infected with HIV; there are reports of cross-neutralization of SARS-CoV-2 by HIV-specific broadly neutralizing Abs [57], suggesting another source of cross-reactive Ab responses against SARS-CoV-2 that could be playing a role in asymptomatic infection and limited disease.

Due to limited availability at the time of the study, only five individuals received any form of SARS-CoV-2 vaccination. Four out of the five individuals who were vaccinated by either the Astra-Zeneca or the Janssen vaccines were cancer patients with HIV co-infection and elicited >50% neutralization activity against SARS-CoV-2. However, all enrolled subjects were asymptomatic with no history of severe COVID-19 disease at the time of enrollment. This included HIV-co-infected cancer patients who were hypothesized to have high rates of SARS-CoV-2 infection and disease because of immunosuppression caused by cancer and cancer therapies [58,59,60]. Prior infection with seasonal CoVs may have resulted in enhanced recognition of SARS-CoV-2 epitopes, and faster and greater responses against SARS-CoV-2, leading to asymptomatic infection and limited disease in both cancer and non-cancer individuals. Several studies have shown immune reactivity to SARS-CoV-2 epitopes or antigens in samples collected before the COVID-19 pandemic suggesting that SARS-CoV-2 cross-reactive immune responses may be derived from non-SARS-CoV-2 antigens [16,19,20]. Importantly, cross-reactive T and B cell epitopes have been computationally predicted and validated to be in the regions of high homology between SARS-CoV-2 and HCoVs, particularly the nucleocapsid protein, the S2 region of the spike protein, and the NSPs. The extent to which these cross-reactive Ab responses are neutralizing and cross-protective against the SARS-CoV-2 spike and preventative against severe COVID-19 remains controversial. Some studies have reported that individuals who were PCR-positive for seasonal CoVs and later became infected with SARS-CoV-2 reported a milder COVID-19 disease compared to those with no prior seasonal CoV infection [61,62]. Similarly, others have reported a less severe COVID-19 disease and a reduced risk of ICU admission and mortality for individuals with histories of seasonal CoV infection [61]. However, other studies have reported that baseline anti-seasonal CoV Ab titers showed no association with the prevention of infection, prediction of COVID-19 severity, or hospitalization [63]. In this cohort, the rate of BCG vaccination was high (75%), and this has been previously linked with protection against severe COVID-19 [64,65], a finding not universally reproduced.

While the peptides presented in the VirScan phage display are often long enough to achieve some secondary structure, the detection of responses against conformational or post-translationally modified epitopes is not supported. This is noteworthy because it is known that the primary target for SARS-CoV-2 neutralization is the receptor-binding domain, which displays a post-translationally modified complex secondary structure. Although we previously reported PhIP-seq-defined Ab responses against non-RBD regions that have been associated with high neutralization activity, those samples were from non-cancer, SARS-CoV-2 infection naïve, vaccinees who were followed from baseline through serial mRNA vaccinations (three rounds). The non-RBD spike responses, similar to those detected in longitudinal vaccination, were not differential between cancer and non-cancer patients in this study, despite evidence of a higher rate of neutralization activity in cancer patients. Thus, it is possible that the nAb activity in the Tanzanian cancer patients was targeting conformational or post-translationally modified epitopes in spike, whereas the responses against linear/quasi-linear epitopes were not differential.

Using phage display, we observed that the CoV Ab repertoire was not significantly different between Tanzanians with and without cancer. However, the segregation of cancer patients into KS and non-KS individuals revealed significant differences in CoV repertoire, HCoV repertoire, and SARS-CoV-2 Ab breadth. Despite the observed differences, the lack of some clinical information and the overall small sample size, particularly for the non-KS cancer individuals, has limited our ability to reveal potential associations. In particular, the undocumented CD4 counts, plasma HIV viral loads, and duration of ART use/HIV infection in this cohort further limited our ability to fully investigate the role of HIV in the anti-SARS-CoV-2 responses. Furthermore, the lack of severe symptoms prevented the comparison of Ab repertoire differentials based on disease severity. Whether symptomatic cancer patients succumbed at home or in the hospital could not be established in this study, although mortality due to COVID-19 has generally been low, including for cancer patients in this region. Moreover, although we implicated a possible cross-protective role of seasonal CoV Ab responses, we could not assess the neutralization capacity against each HCoV, or even all nAb responses against SARS-CoV-2, due to the limited quantity of sera collected. Despite these limitations, our study has revealed a positive association between seasonal CoV and the magnitude of the anti-SARS-CoV-2 Ab responses both in individuals with cancer (KS) and those without cancer. Seasonal HCoV infections may have played an important role in limiting infection and disease among vulnerable Tanzanian cancer patients. However, further studies are needed to clarify the protective role of anti-HCoV humoral responses in limiting SARS-CoV-2 and COVID-19 severity among cancer patients.

## 5. Conclusions

By using PhIP-seq technology to investigate the role of prior HCoV infection in anti-SARs-CoV-2 responses, it appeared that SARS-CoV-2 epitopes were the most consistently recognized Ab repertoires. Despite the lower magnitude of anti-SARS-CoV-2 Ab responses in individuals with KS compared to non-KS and non-cancer individuals, there was a strong association with HCoVs, suggesting that prior infection with HCoVs might play a role in limiting SARS-CoV-2 infection and COVID-19 disease severity.

## Figures and Tables

**Figure 1 viruses-16-00448-f001:**
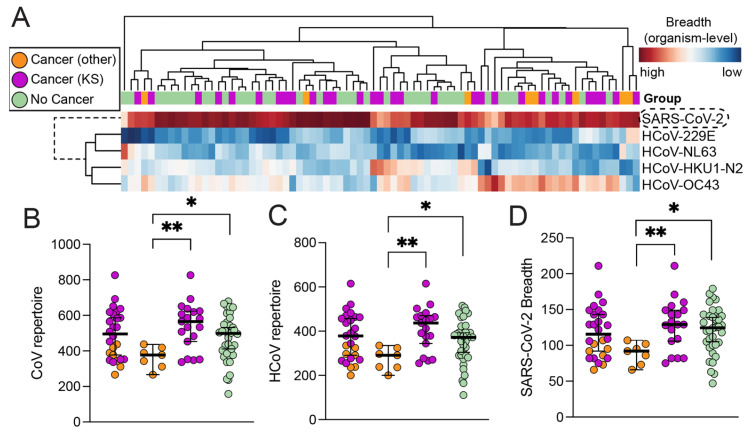
**Comparison of quantified Ab responses.** (**A**) Comparison of breadth across organisms. SARS-CoV-2 breadth was consistently higher across individuals compared to other seasonal CoVs and was not associated with age, gender, or HIV/disease status. Hierarchical clustering (McQuitty + Euclidean) (**B**) combined breadth of responses against SARS-CoV-2 and all seasonal human coronaviruses (i.e., CoV repertoire), separated into (**C**) HCoV repertoire only, and (**D**) SARS-CoV-2 breadth. Similar observations of each level can be made between cancer (n = 26) and non-cancer (n = 36) patients. Interestingly, anti-CoV/HCoV/SCoV-2 breadth is significantly higher in KS patients (n = 19) compared to in patients with other cancer types (n = 7). Abbreviations for *p*-values: ** *p* < 0.01, * *p* < 0.05.

**Figure 2 viruses-16-00448-f002:**
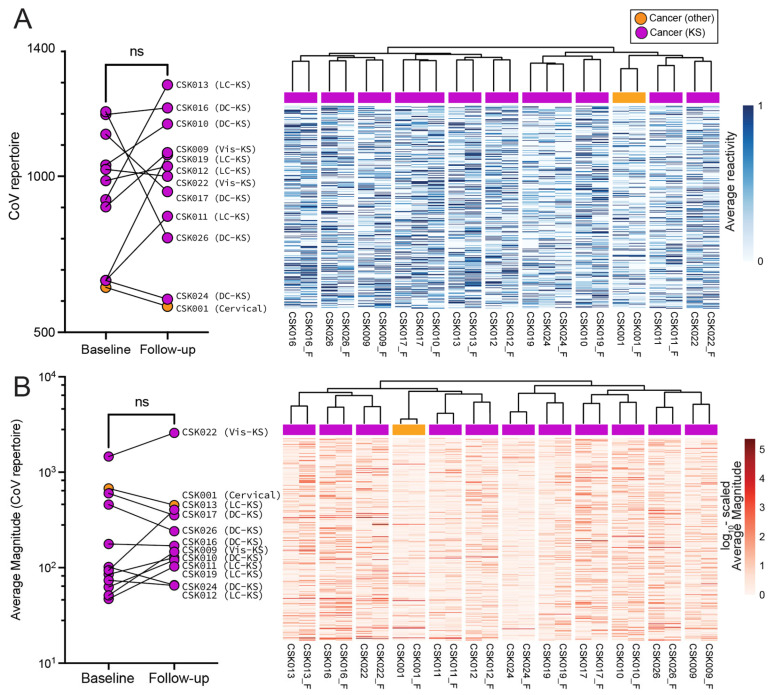
**Comparison of the changes in Ab responses between baseline and follow-up cancer patients (n = 12).** (**A**) (Left) statistical and (Right) hierarchical clustering of patients with baseline and follow-up timepoints (indicated with _F) based on peptide reactivity against all CoV organisms. Cluster hierarchy indicated that reactivity patterns are highly patient-specific, and not influenced by cancer treatment regimens. Follow-up patients without cancer do not form a cluster that is distinct from patients with cancer and are also individualistic in overall anti-CoV reactivity patterns. (**B**) Similar observations were made when magnitudes of responses were compared. Statistical comparisons of baseline-follow-up pairs were conducted by a Wilcoxon matched-pairs signed rank test.

**Figure 3 viruses-16-00448-f003:**
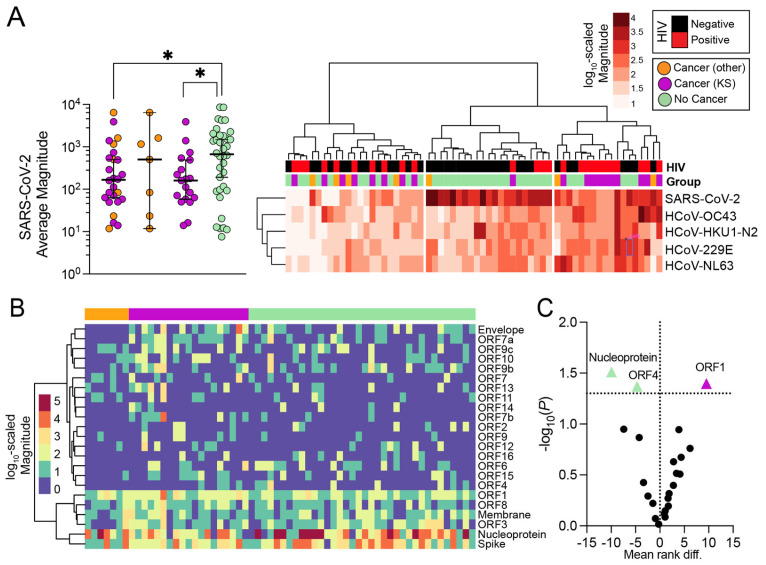
**Organism- and protein-level magnitude of responses.** (**A**) Comparison of overall SARS-CoV-2 magnitudes between groups (**left**), as well as hierarchical clustering analysis of patients given magnitudes of each HCoV organism with SARS-CoV-2 (**right**). (**B**) Comparison of SARS-CoV-2 specific protein-level magnitude of responses. (**C**) Statistical comparison of each protein from (**B**) between cancer and non-cancer individuals (Mann–Whitney test, without multiple-hypothesis testing). Abbreviations for *p*-values: * *p* < 0.05.

**Figure 4 viruses-16-00448-f004:**
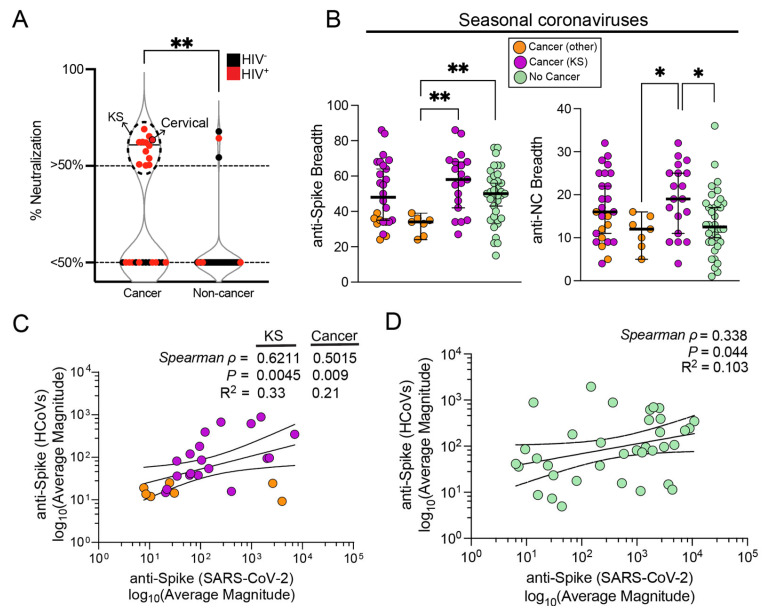
**HCoV repertoire and neutralization in association with anti-SARS-CoV-2 responses.** (**A**) Comparison of neutralization activity between cancer and non-cancer participants, where HIV co-infected individuals are shown in red. (**B**) Protein-level HCoV breadth comparison among cancer and non-cancer individuals, focusing on anti-spike and anti-nucleocapsid (NC) responses. Lastly, regression and correlation analyses of anti-spike SARS-CoV-2 and HCoV magnitudes in (**C**) cancer and (**D**) non-cancer groups. Abbreviations for *p*-values: ** *p* < 0.01, * *p* < 0.05.

**Table 1 viruses-16-00448-t001:** **Sociodemographic and clinical characteristics of the study participants.** Abbreviations; IQR: Inter-quartile range, BCG: Bacillus Calmette–Guérin. Mann–Whitney U and Fisher’s exact tests were used for continuous and categorical variables, respectively. Significance threshold was set at α = 0.05.

Variable	All (N = 62)	Cancer (n = 26)	Non-Cancer (n = 36)	*p*-Value
**Age|**Median (IQR), years	44 (30.25)	38 (16.5)	55.5 (33.75)	**0.0027**
**Sex|**n (%)				0.4425
Female	28 (45.2)	10 (38.5)	18 (50)	
Male	34 (54.8)	16 (61.5)	18 (50)	
**HIV Status|**n (%)				**<0.0001**
Negative	37 (59.7)	7 (29.9)	30 (83.3)	
Positive	25 (40.3)	19 (73.1)	6 (16.7)	
**Vaccinated TB BCG|**n (%)				0.5501
Yes	46 (75.4)	21 (80.8)	25 (71.4)	
No	15 (24.6)	5 (19.2)	10 (28.6)	
**Unknown**	1 (1.6)	0	1 (2.8)	
**Vaccinated SARS-CoV-2|**n (%)				0.1516
Yes	5 (8.1)	4 (15.4)	1 (2.8)	
No	57 (91.9)	22 (84.6)	35 (97.2)	
**COVID-19 Like Symptoms|**n (%)				0.0874
**Yes**	18 (29.0)	11 (42.3)	7 (19.4)	
**Cough**	6 (9.7)	5 (19.2)	1 (2.8)	
**DIB**	1 (1.6)	1 (3.9)	0	
**Cough and DIB**	2 (2/62)	1 (3.9)	1 (2.8)	
**Fever**	8 (12.9)	4 (15.4)	4 (11.1)	
**Fever and cough**	1 (1.6)	0	1 (2.8)	
**No**	44 (71)	15 (57.7)	29 (80.5)	

## Data Availability

The data presented in this study are available upon request.

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
