# Peer review of "Viral Epitope Scanning Reveals Correlation between Seasonal HCoVs and SARS-CoV-2 Antibody Responses among Cancer and Non-Cancer Patients"

_viruses, 2024, doi:10.3390/v16030448_

Round 1
Reviewer 1 Report
Comments and Suggestions for Authors
Lidenge et al. present interesting data on seasonal vs. SARS-CoV-2 coronaviruses in a cohort of patients from Africa.
Their demonstrated findings of correlation between anti-hCoVs responses and anti-SARS-CoV-2 responses could be added to the global knowledge of these viruses, however I'm not convinced by the conclusions that they draw overall from their data, as well as by the choice of groups for data analysis.
There are significant differences demonstrated between the responses of KS vs. non-KS cancer patients, as well as KS vs. non-cancer patients in some parameters, as well as between non-KS cancer vs. non-cancer patients in others. The lack of consistentcy in the observed differences precludes, in my opinion, generalization of conclusions to "cancer" vs. "non-cancer" patients, and should instead be directed to conclusions specific to KS vs. other cancers and/or non-cancer patients.
Furthermore, while the authors state that there were no significant influences observed of HIV status on quantified and qualified serological responses, these comparisons are not actually provided to the reader, but it would be good to provide them, in particular given the significantly uneven distribution of HIV diagnoses between the groups. Similarly, the "HIV status" of individuals is only qualified by the fact that they are on HAART and in the discussion a "high rate of adherence" is mentioned. But there's no information on CD4 status, virological suppression, age at diagnosis, duration of illness prior to HAART initiation, etc. Given the potential for high heterogeneity of HIV outcomes and impact of HIV infection on overall immune status depending on all these parameters, it would be essential to provide this data on the study participants, as well as attempt to analyze their serological responses by stratifying them based on CD4 status and/or similar markers of immune status, in addition to just "HIV+" or "HIV-" status.
Some more detailed comments:
p.2 lines 49-50 - provided references discuss transmission of SARS-CoV-2 - to support the assertion the authors make there should be references that discuss transmission of hCoVs as well.
p.2 line 90 - ADCC is not defined appropriately (should be spelled out as antibody-dependent cellular cytotoxicity)
p.5 line 242 - asymptomatic at the time of recruitment or ever since the start of the pandemic? Knowing about symptoms that could have been attributable to COVID-19 in the preceding months/years could help time the exposure of the participants.
p.7 line 282/Figure 1A - did you do any evaluations of OC43+/- HU1 relative to other hCoVs and OC43 individually relative to SARS-CoV-2? - it's the most related betacoronavirus and both betacoronaviruses look "redder" on the graph relative to alpha hCoVs
p. 13 lines 463-464 - differences between cancer types (KS and other) are actually well demonstrated by your work, contrary to what is stated in the discussion.
Author Response
Please see the word document attached, for the responses to Reviewer 1.

Reviewer 2 Report
Comments and Suggestions for Authors
Revision manuscript “Viral epitope scanning reveals correlation between seasonal HCoVs and SARS-CoV-2 antibody responses among cancer patients”
In this small study, the authors investigated the antibody response in cancer and non-cancer patients versus SARS-CoV-2 and seasonal HCoVs. They found that seasonal HCoV infection seems correlate with the magnitude of anti-SARS-CoV-2 Spike in cancer patients suggesting that previous infection with seasonal coronaviruses might limit SARS-CoV-2 and COVID-19 disease severity in cancer patients. A possible cross-reaction of the antibodies against seasonal coronaviruses could limit SARS-CoV-2 infection.
The manuscript is interesting even though the hypothesis of the authors needs validation on a larger sample size, and the neutralizing activity of the antibodies should be tested versus the single seasonal coronaviruses and SARS-CoV-2.
The authors report, lines 345-347: “Anti-NC, and anti-ORF4 magnitudes were, on average, significantly higher in non-cancer individuals, while anti-ORF1 magnitude were significantly higher in cancer patients (p=0.031, p=0.043, and p=0.040, respectively”. How do you interpret these results?

Author Response
Please see the word document attached, for the responses to Reviewer 2.

Reviewer 3 Report
Comments and Suggestions for Authors
The present study aimed to investigate the influence of prior HCoVs infection on 20 anti-SARS-CoV-2 Ab responses among COVID-19 asymptomatic individuals with cancer and controls without cancers.
The article is very interesting.
The images are profession and well designed.
Plsea try not to cite so many articles from your group.
In the discussion try to cite articles that are relevant for your study.
Also perform an english editing.
Please include also a paragraph of conclusion.
Comments on the Quality of English LanguageModerate
Author Response
Please see the Word document attached, for the responses to Reviewer 3.

Round 2
Reviewer 1 Report
Comments and Suggestions for Authors
The authors explained in their response to reviewers that they did not have access to supplementary information on their participants to conduct the additional recommended analyses.
I understand that it can be difficult/impossible to obtain all relevant information, but in the absence of it the data analyses must still be conducted in a clinically sound manner, based on the available information. Conducting the analyses in the current fashion, without accounting for HIV-related immunosuppression of their participants, does not seem sound. Given the lack of clinical information on HIV patients, it would make much more sense to conduct analyses on HIV+/cancer vs. HIV-/cancer vs. HIV+/non-cancer vs. HIV-/non-cancer groups rather than mixing them up.
Additionally, some of the new references the authors provided in the introduction that are meant to support the differences in the transmission mechanisms of hCoVs vs. SARS-CoV-2 don't address this topic at all.